# Genome- and Transcriptome-Wide Characterization of *bZIP* Gene Family Identifies Potential Members Involved in Abiotic Stress Response and Anthocyanin Biosynthesis in Radish (*Raphanus sativus* L.)

**DOI:** 10.3390/ijms20246334

**Published:** 2019-12-16

**Authors:** Lianxue Fan, Liang Xu, Yan Wang, Mingjia Tang, Liwang Liu

**Affiliations:** National Key Laboratory of Crop Genetics and Germplasm Enhancement, Key Laboratory of Horticultural Crop Biology and Genetic Improvement (East China) of MOAR, College of Horticulture, Nanjing Agricultural University, Nanjing 210095, China; 2016204023@njau.edu.cn (L.F.); nauxuliang@njau.edu.cn (L.X.); wangyanhs@njau.edu.cn (Y.W.); 2017204022@njau.edu.cn (M.T.)

**Keywords:** *bZIP* gene, *Raphanus sativus* L., abiotic stress, anthocyanin biosynthesis, gene expression

## Abstract

Basic leucine zipper (bZIP) transcription factors play crucial roles in various abiotic stress responses as well as anthocyanin accumulation. Anthocyanins are most abundant in colorful skin radish, which exhibit strong antioxidant activity that offers benefits for human health. Here, a total of 135 bZIP-encoding genes were identified from radish genome. Synteny analysis showed that 104 radish and 63 *Arabidopsis*
*bZIP* genes were orthologous. Transcriptome analysis revealed that 10 *RsbZIP* genes exhibited high-expression levels in radish taproot (RPKM>10). Specifically, *RsbZIP010* exhibited down-regulated expression under Cd, Cr and Pb stresses, whereas *RsbZIP031* and *RsbZIP059* showed significant down-regulation under heat and salt stresses, respectively. RT-qPCR analysis indicated that *RsbZIP011* and *RsbZIP102* were significantly up-regulated in the tissues of radish with high anthocyanin contents. Furthermore, the promoter sequences of 39 anthocyanin-related genes were found to contain G-box or ACE-box elements that could be recognized by bZIP family members. Taken together, several *RsbZIP*s might be served as critical regulators in radish taproot under Cd, Cr, Pb, heat and salt stresses. *RsbZIP011* and *RsbZIP102* were the potential participants in anthocyanin biosynthesis pathway of radish. These results facilitate further investigation on functional characterization of *bZIP* genes in response to abiotic stress and anthocyanin biosynthesis in radish.

## 1. Introduction

The basic leucine zipper (bZIP) family, as one of the largest and most diverse transcription factor (TF) families, is an important participator in various biological processes, such as seed maturation and germination, biotic or abiotic stress responses and hormone signaling [1,2]. It is widely known that bZIP TF possesses a conserved bZIP domain composed of two structural features, a DNA-binding basic region and a dimerizing leucine (Leu) zipper region [1,3]. The basic region of approximately 16 amino acids (aa) is highly conserved containing a nuclear localization signal followed by an invariant N-X_7_-R/K motif that binds to specific DNA sequences with an ACGT core, such as A-box (TACGTA), C-box (GACGTC) and G-box (CACGTG) [4]. The Leu zipper has a less conserved dimerization motif and is composed of a heptad repeat of Leu or other similar hydrophobic aa (e.g., Ile, Met, Phe or Val) positioned exactly nine aa towards the C-terminus, creating an amphipathic helix [5,6]. To bind with DNA, two subunits adhere via interactions between the hydrophobic sides of their helices, which creates a superimposing coiled-coil structure for recognition and dimerization [2,5].

The bZIP TF family has been extensively identified in various plant species, such as *Arabidopsis*, rice, rapeseed, tomato and apple [1,7,8,9,10]. The *Arabidopsis bZIP* gene family consisted of 78 members are assorted into 13 groups (Groups A–K, M and S) [1]. Recently, several *bZIP*s were found to be involved in response to a wide range of abiotic stresses, such as heat and high-salinity stresses [11,12]. Interestingly, increasing evidences indicate that a few *bZIP* genes such as *HY5* (*Elongated hypocotyl 5*) and *HYH* (*HY5 homolog*) are involved in the biosynthesis pathway of anthocyanins in plants [13,14,15]. In *Arabidopsis*, *HY5* can directly bind the promoters of some *MYB TF* genes, or together with other MYB factors, induce the expression of biosynthetic enzyme genes associated with PHYTOCHROME INTERACTING FACTOR3 activity for the promotion of anthocyanin accumulation [14]. Similarly, *HY5* has been found to play a vital role in anthocyanin biosynthesis in tomato and apple [13,15]. These results provide useful information for comprehensive identification and characterization of *bZIP*s in root vegetable crops.

Radish (*Raphanus sativus* L.), belonging to the Brassicaceae family, is a major root vegetable crop with plenty of nutrition, and its taproot is an edible and stable food supplying a great source of phytochemicals such as anthocyanins [16,17]. Owning to the rapidly changeable climate and environment, radish growth and production are often severely limited by various abiotic stresses such as high temperature, salinity or metal pollution [18,19,20,21,22,23]. Although some bZIP TFs have been proven to be associated with different abiotic stress responses in many plant species, little information on the characterization of bZIP family is available in radish. The first draft sequences of the radish genome were released into the Raphanus sativus Genome DataBase (http://radish.kazusa.or.jp/) in 2014 [16], and its available genomic sequences and proteins/mRNA transcriptomics data provided valuable information for systematic identification of functional genes and TFs associated with important horticultural traits in radish. In this study, to identify the candidate genes that may participate in abiotic stress response and anthocyanin biosynthesis in radish, the *bZIP* gene family members were systematically identified at the genome-wide level. Herein, a total of 135 *bZIP* gene members were identified based on the radish genome sequences, and the chromosome distribution, gene structure and conserved motifs were further investigated. Moreover, the expression patterns of *RsbZIP* genes were analyzed in different tissues, and seven *RsbZIP* genes that exhibited specific expression with high-levels in taproot were explored as the possible regulators in response to various stresses conditions such as cadmium (Cd), chromium (Cr), lead (Pb), heat and salt stresses. Furthermore, two potential *bZIP* genes related to anthocyanin biosynthesis were firstly identified in radish. These results could provide valuable information on evolutionary relationship and functional divergence of the *bZIP* gene family in radish.

## 2. Results

### 2.1. Identification and Chromosome Distribution of bZIP Family in Radish

Using the HMM profile analysis, a total of 137 protein sequences were identified containing bZIP domains from the radish genome. Further SMART and InterPro analyses revealed that 135 genes could be assigned as radish *bZIP* genes, which were named *RsbZIP001*~*RsbZIP135* accordingly (Appendix A). Among them, 128 of 135 *RsbZIP* members were localized on the nine chromosomes (Chrs) of radish (Figure 1a,b). To be specific, Chr6 contained the highest number of *RsbZIP* genes (31, 23%), followed by Chr4 (25, 19%), while minimum genes were distributed on Chr1 (6, 4%) (Figure 1a). The tandem and segmental duplication events were also investigated. In all, only three pairs of tandem duplicates in *RsbZIP* gene family were determined, including *RsbZIP005*-*RsbZIP006*, *RsbZIP037*-*RsbZIP038* and *RsbZIP085*-*RsbZIP086* (Figure 1b). Moreover, 27 (20%) segmental duplicated *RsbZIP* genes were detected to localize on two different chromosomes (Appendix A).

### 2.2. Gene Structure and Conserved Motif Analysis of RsbZIPs

Exon-intron organizations of the 135 *RsbZIP*s were investigated to obtain the gene structure and evolutionary trajectory. The number of introns varied from 0 to 12 in *RsbZIP* genes and the genes that clustered together generally possessed similar gene structures (Figure 2a,b). A total of 37 (27%) *RsbZIP*s with no intron were detected, including 35 from Group S and two from Group A. Groups B, C, E, F, H, J and M had 1–6 introns, whereas the *RsbZIP* genes containing a minimum of 7 introns appeared in Groups D, G and I (Figure 2a,b).

A total of 20 conserved motifs were identified from all the RsbZIP proteins. Interestingly, the RsbZIPs within the same group exhibit similar motif compositions, which further supported the group classification. Motif 1 was the basic region and the hinge of bZIP domain, which existed in all 135 RsbZIP proteins. Motifs 2, 5, 8 and 10 represented the variable motifs in the Leu zipper region across the radish bZIP family (Figure 2c, Appendix A). For example, Motif 5 appeared in Groups I and E, and Motif 8 appeared in Group D, while the Motif 10 only appeared in Groups S and C (Figure 2a,c). The number of Leu units ranged from 3 to 7, and some of them were interrupted by one or two other units which are conserved with the same number of aa such as RsbZIP044 (Appendix A). Several additional motifs were also identified in some groups. For instance, most members from Group A shared Motifs 9, 11, 12, 18 and 20, while Group B possessed Motifs 3, 16 and 19. The members from Group D harbored Motifs 3, 4, 6 and 7, while Group G had Motifs 15 and 16, and Group I shared Motifs 14 and 17. The Group E possessed Motif 13 exclusively, whereas Groups C, F, H, J and M lacked any additional motif (Figure 2a,c). These specific motifs might imply diverse functions of the bZIP family in radish.

### 2.3. Phylogenetic and Evolutionary of Radish and Arabidopsis bZIP Genes

To investigate the phylogenetic relationship of the *bZIP* gene families, the aa sequences encoded by the *bZIP* genes from radish and *Arabidopsis* were used to construct an unrooted Neighbor-Joining (NJ) tree (Figure 3, Appendix A). As shown in Figure 3, the 213 bZIP proteins could be divided into 13 subfamilies, which were named as Groups A, B, C, D, E, F, G, H, I, J, K, M and S according to the *Arabidopsis* classification [1]. Group S was the largest subfamily, consisting of 37 RsbZIP members, followed by the Groups A and I containing 23 and 18 RsbZIP proteins, respectively. The 16 and 10 RsbZIP members were classified into Groups D and E, while the number of members in Groups B, H, F, C and G was four, seven, four, six and eight, respectively. Both Groups J and M only had one RsbZIP protein, respectively. Interestingly, the Group K included only AtbZIP60 without any RsbZIP member, indicating that this group might be specific to *Arabidopsis* (Figure 3). According to the functional and regulatory aspects of the bZIP subgroups in *Arabidopsis*, the classification of *RsbZIP* gene family might serve as their phylogeny-based functional prediction.

To deduce the origins and evolutionary history of *bZIP* genes, a comparative bZIP synteny map was conducted between radish and *Arabidopsis*. Large-scale syntenies showed that 104 radish and 63 *Arabidopsis bZIP* genes were orthologous (Figure 4, Appendix A). Among them, 12 pairs of syntenic orthologous genes (one to one) were identified, such as *RsbZIP003*-*AtbZIP62*, *RsbZIP050*-*AtbZIP21* and *RsbZIP056*-*AtbZIP51*. These genes may be derived from a common ancestor of radish and *Arabidopsis*. Two syntenic orthologous gene pairs with one radish gene corresponding to multiple *Arabidopsis* genes were also identified including *RsbZIP087*-*AtbZIP16/68* and *RsbZIP094*-*AtbZIP54/55*. Correspondingly, there also existed syntenic orthologous gene pairs with one *Arabidopsis* gene corresponding to multiple radish genes, *AtbZIP01*-*RsbZIP032/123*, *AtbZIP09*-*RsbZIP030/071* and *AtbZIP41*-*RsbZIP014/113*. Additionally, the gene pairs where two *Arabidopsis* genes correspond to the same (two, three or four) radish genes were also found, such as *AtbZIP13/40*-*RsbZIP036/116*, *AtbZIP22/50*-*RsbZIP106/112/118* and *AtbZIP59/69*-*RsbZIP001/039/085/110* (Figure 4, Appendix A). A range of synteny events suggested that many *bZIP* genes arose before the divergence of the *Arabidopsis* and radish lineages.

### 2.4. Expression Profiles of bZIP Genes in Different Tissues of Radish

To investigate the tissue-specific expression profiles of *bZIP* genes in different developmental stages of radish, an expression analysis of *RsbZIP* genes in five tissues (leaf, cortical, cambium, xylem and root tip) were displayed with RNA-seq data from Mitsui et al. [18]. A heatmap was conducted based on the reads per kilobase per million reads (RPKM) values of 135 *RsbZIP* genes (Figure 5a). All *RsbZIP* genes exhibited very diverse expression profiles in five tissues. On average, the expression of 68%, 61%, 57%, 51% and 47% *RsbZIP* genes were detected in these five different tissues, when RPKM values were greater than 2, 4, 6, 8 and 10, respectively (Figure 5b, Appendix A). In detail, 39%, 47%, 45%, 50% and 51% *RsbZIP* genes showed higher transcriptional abundance (RPKM value>10) in leaf, cortical, cambium, xylem and root tip, respectively (Figure 5b). Among them, 41 *RsbZIP* genes including *RsbZIP001/004/009* were abundantly expressed (RPKM value>10) in all tissues (Figure 5c,d), while 31 genes (e.g., *RsbZIP007/008/016*) were expressed at relatively low levels. Moreover, a total of 10 *RsbZIP*s (e.g., *RsbZIP031/059/064*) were selectively expressed with high-levels (RPKM value>10) in taproot (cortical, cambium, xylem and root tip), while four *RsbZIP*s (e.g., *RsbZIP047/063/069*) were only in leaf with high-level expression (Figure 5a). These *RsbZIP* genes might play important roles in radish growth and development process.

### 2.5. Potential Roles of RsbZIP Genes in Various Abiotic Stress Responses

The expression patterns of *RsbZIP* genes under Cd, Cr, Pb, heat and salt stresses were investigated in radish according to previously root transcriptome data [19,20,21,22,23]. As shown in Figure 6a, a total of 76, 77, 38, 43 and 46 *RsbZIP* genes were differentially expressed under Cd, Cr, Pb, heat and salt stresses (the absolute value of Fold change > 1, *p*-value < 0.05), respectively. To provide more evidences for the *RsbZIP* genes’ roles in response to different abiotic stresses, the expression of the 10 *RsbZIP* genes (*RsbZIP005/006/010/031/059/064/076/077/092/101*), which were selectively expressed at high-levels in taproot, was analyzed under Cd, Cr, Pb, heat and salt stresses. As shown in Figure 6b–f, seven of these 10 *RsbZIP* genes exhibited significantly differential expression in taproot under the Cd, Cr, Pb, heat and salt treatments. For example, the *RsbZIP010* gene exhibited down-regulated expression under three heavy metal (HM) stresses, whereas the *RsbZIP005* gene was significantly up- and down-regulated under Cd and Cr stress, respectively (Figure 6b–d), indicating that the *RsbZIP010* gene might play conserved key roles in the biological process of response to three HM stresses in radish taproot. Notably, the *RsbZIP031* and *RsbZIP059* genes showed significant down-regulation under heat and salt stresses, respectively (Figure 6e,f). This finding indicated that these two genes may act as critical regulators in heat and salt stress response in radish taproot. However, there was no expression of *RsbZIP076*, *RsbZIP077* and *RsbZIP101* in taproot under the five abiotic stresses. To validate the expression profiles of *RsbZIP* genes obtained from the RNA-seq data, seven high-expressed *RsbZIP* (*RsbZIP005/006/010/031/059/064/092*) genes in taproot combined with another six *RsbZIP* genes were selected for real-time quantitative polymerase chain reaction (RT-qPCR) analysis. The expression patterns from RT-qPCR indicated a general agreement with those from the RNA-seq results (Figure 6b–f, Appendix A).

### 2.6. Potential Roles of RsbZIP Genes in Anthocyanin Biosynthetic Pathway

Two Group H members of *Arabidopsis* bZIP family, *HY5* (*AtbZIP56*) and *HYH* (*AtbZIP64*), can be acted as important participants in the process of anthocyanin accumulation [1,14]. In this study, seven radish *bZIP* genes (*RsbZIP011/028/063/065/082/097/102*) that were homologous to *Arabidopsis HY5* and *HYH* were used for expression profiles in ten radish genotypes with red-skin and white-skin using RT-qPCR. The results showed that the expression levels of *RsbZIP011* and *RsbZIP102* were significantly up-regulated in five red-skin genotypes compared with that in five white-skin ones (Figure 7a,b, Appendix A). Furthermore, *RsbZIP011* and *RsbZIP102* also exhibited higher expression levels in five tissues (root skin, stem, leaf vein, septal and petal) with high anthocyanin contents than that in root flesh with low anthocyanin content (Figure 7c–e), indicating that *RsbZIP011* and *RsbZIP102* may participate in anthocyanin biosynthetic pathway in radish.

## 3. Discussion

### 3.1. Characterization of bZIP Gene Family Members in Radish

The *bZIP* gene family, one of the largest TF families, has been extensively characterized in many plants, including 78 *bZIP* genes in *Arabidopsis* [5], 69 in tomato [7], 89 in rice [8] and 247 in rapeseed [10]. In this study, 135 *bZIP* genes were identified from the radish genome, indicating that the *bZIP* gene family in radish had been expanded compared to that in *Arabidopsis*, rice and tomato. Phylogenetic analysis indicated that the radish bZIP family could be divided into 12 subfamilies based on cluster analysis and a comparison with *Arabidopsis bZIP* genes. The phylogenetic relationship of RsbZIPs was also supported by both their gene structure and conserved motif.

Gene structure analysis showed that *RsbZIP* genes possessed intron with number varying from 0 to 12, and each subfamily displayed similar exon-intron organizations. In total, 44 *RsbZIP* genes contained lower intron number (0 or 1 intron), which is similar in soybean [4], apple [9] and rapeseed [10]. Consistent with the results in soybean and rapeseed, the *RsbZIP* genes from Groups D and G contained more than six introns, suggesting that they might contain the original *RsbZIP* genes in these groups [4,10]. In this study, exon/intron gain/loss was also observed. For instance, *RsZIP050* contained 13 exons, while its paralogous *RsbZIP135* had 11 exons (Figure 2), indicating a loss of two exons occurred during evolution. A similar pattern was also reported in the apple and soybean bZIP families [4,9]. These gain/losses may be the results of chromosomal rearrangements and fusions, and can potentially lead to the functional diversification of multiple gene families [24]. Additionally, two genes in a duplicated gene pair tended to be clustered into one group. The tandem duplicated pairs *RsbZIP005*-*RsbZIP006* and *RsbZIP037*-*RsbZIP038* were clustered into Groups A and I, respectively. The segmental pairs *RsbZIP002*-*RsbZIP078* and *RsbZIP106*-*RsbZIP118* were divided into Groups S and D, respectively. These duplicated genes had similar exon-intron organizations or nearly identical lengths, indicating that segmental or tandem duplication might contribute to the expansion of *RsbZIP* gene family [4].

Conserved motif analysis showed that all the RsbZIP proteins shared the typical bZIP domain (Motif 1) and variable Leu zipper regions (Motifs 2, 5, 8 and 10), and each subfamily harbored similar motifs. Motifs 12 and 20 shared the same domain such as the potential casein kinase II phosphorylation sites (S/TxxD/E), which were contained only in the Group A. Motifs 15 and 16 in Group G were part of a proline-rich activation domain. These findings indicated that some RsbZIP subfamily members might be conserved. In summary, the same subfamily tends to share similar exon-intron organization and motif composition, which is also reported in rapeseed [10].

### 3.2. Collinearity-Orthologues of the bZIP Genes between Radish and Arabidopsis

Comparative genomic analysis across different taxa relies on the genome structure into syntenic blocks that display conserved features across the genomes [25], which allows the transfer of functional information from a well-studied taxon (e.g., the model plant *Arabidopsis*) to the less-studied taxon. In this study, the synteny analysis integrating with evolutionary classification was performed to assess the relationship of *bZIP* genes between radish and *Arabidopsis*. A total of 104 radish and 63 *Arabidopsis bZIP* genes were identified as syntenic orthologs. Of these, nine pairs appeared to be single radish-to-*Arabidopsis* pairs, indicating that these genes likely present in the genome of the last common ancestor of these two species. The remaining gene combinations showed a more complex relationship, such as single radish-to-multiple *Arabidopsis* genes, one *Arabidopsis*-to-multiple radish genes and two *Arabidopsis*-to-multiple radish genes.

In addition, a small proportion of the *bZIP* genes from these two species could not be mapped into any syntenic blocks, leading into impossibility to determine whether these *bZIP*s share a common ancestor. This phenomenon may be attributed to two factors, which is also true for the collinearity analysis between *Arabidopsis* and other species such as apple [9]. Firstly, the radish and *Arabidopsis* genomes have undergone multiple rounds of significant chromosomal rearrangement and fusions. Secondly, selective gene loss can severely obscure the identification of chromosomal syntenies. It could be concluded that some radish *bZIP* genes tend to share a common ancestor with their *Arabidopsis* counter parts. Moreover, *AtbZIP01* and its partner *AtbZIP53* control metabolic reprogramming upon salt stress in roots [26], while one of their orthologues, *RsbZIP123* was also highly inducible under salt stress in this study (Appendix A). Collectively, comparative genomic analysis between radish and *Arabidopsis* would serve as a valuable reference for further investigation of the biological functions of *RsbZIP* genes.

### 3.3. Roles of RsbZIP Genes in Different Tissues and Abiotic Stress Responses

Increasing evidences indicate that the *bZIP* genes can act as key components under a wide range of abiotic stresses including extreme temperature and high salinity [11,12]. The transcriptome data showed that all 135 *RsbZIP* genes exhibited diverse expression profiles in different tissues. Nearly a third (41/135) of *RsbZIP* genes had high transcriptional abundance in all tissues, indicating that they may play a broad role in radish growth and development. Ten *RsbZIP*s were specific high-expression genes in radish taproot, suggesting their potential functions during the development of radish taproot. Expression profiling based on RNA-seq data and RT-qPCR analysis further showed that seven of these 10 *RsbZIP* genes exhibited significantly differential expression in taproot under the Cd, Cr, Pb, heat and salt treatments. In particular, *RsbZIP010* exhibited down-regulated expression under Cd, Cr and Pb stresses, whereas *RsbZIP031* and *RsbZIP059* showed significant down-regulation under heat and salt stresses, respectively. However, *RsbZIP031* was up-regulated under salt and three HM stresses, showing its involvement in response to salt stress and HM detoxification. Moreover, *RsbZIP059* was up-regulated under Pb and Cd stresses. These *RsbZIP* genes might play conserved roles in the biological process of five abiotic stress responses in radish taproot. This finding was also supported by gene ontology (GO) and *cis*-element analyses of *RsbZIP*s (Appendix A). GO annotation showed that 43 RsbZIP proteins were classified into response to stimulus, and stress-related *cis*-elements including the ABRE and ERE were found in the promoter regions of most *RsbZIP* genes, which is consistent with the results observed in other plants including grapevine and soybean [4,6]. These results indicated that the *bZIP* gene might play vital roles in abiotic stress response in radish taproot.

### 3.4. Potential Function of bZIP Genes in Regulation of Anthocyanin Biosynthesis

Anthocyanins are a valuable source of phytochemicals and offer benefits for human health [27,28]. Anthocyanins are known to be synthesized as a protective compound when the plants are exposed to various stressful events, such as low temperature, drought, ultraviolet radiation, high-intensity light and nutrient deficiency [15,29]. In *Arabidopsis*, tomato and apple, *HY5* is a positive regulator during anthocyanin accumulation through direct binding to the G-box (CACGTG) element or ACGT-containing element (ACE) of anthocyanin-related gene promoters [13,14,15]. Here, RT-qPCR analysis further showed that the expression levels of *RsbZIP011* and *RsbZIP102* from Group H were associated with anthocyanin accumulation in radish. The G-box or ACE-box elements were further found in the promoters of 39 genes involved in anthocyanin biosynthesis pathway, such as *Rs065380* (*Chalcone synthase*), *Rs063340* (*Chalcone flavanone isomerase*), *Rs341050* (*Flavanone-3-hydroxylase*), *Rs119880* (*Anthocyanidin synthase*), *Rs442410* (*UDP-glucose flavonoid 3-o-glucosyltransferase*), *Rs358990* (*Phenylalanine ammonia-lyase*) and *Rs131400* (*Cinnamate-4-hydroxylase*) (Appendix A). It is reasonable to hypothesize that some anthocyanin-related genes may be regulated by the *Arabidopsis HY5* homolog genes, such as *RsbZIP011* and *RsbZIP102* in radish.

Additionally, the promoters of two potentially anthocyanin-related genes (*RsbZIP011* and *RsbZIP102*) contained several similar *cis*-acting elements involved in ABRE, MBS and G-box element (Appendix A), indicating that they may be involved in ABA-dependent or independent stress tolerance, and could be regulated by MYB or other bZIP TFs. Thus, they may be related not only to anthocyanin biosynthesis but also to other stress responses. The *Arabidopsis HY5* was found to have multifaceted roles in plant growth and development [14]. Moreover, the biotic or abiotic stressors (such as cold or heat stress) could significantly induce anthocyanin accumulation [30,31]. Therefore, it could be inferred that bZIP members might be served as a linkage between stressor and anthocyanin biosynthesis in radish.

In summary, this study describes the systematic genome- and transcriptome-wide characterization of *bZIP* gene family in radish. The phylogenetic classification, gene structures and conserved motifs of 135 bZIP members were investigated. Transcriptomic and RT-qPCR analysis revealed diverse expression profiles of *RsbZIP* genes in taproot. Notably, the *RsbZIP010* gene exhibited down-regulated expression under Cd, Cr and Pb stresses, indicating that it might play conserved key roles in the biological process of response to three HM stresses in radish taproot. Except the heat stress, the *RsbZIP031* gene was up-regulated under salt and three HM stresses, suggesting it might act as an important regulator in the regulation of salt and HM stress responses. Moreover, it was evidenced that *RsbZIP011* and *RsbZIP102* genes were important potential participants in anthocyanin biosynthesis pathway in radish. Further functional characterization of these *RsbZIP* genes would be useful for systematical elucidation of the molecular mechanism underlying the abiotic stress response and anthocyanin biosynthesis in radish plants. The outcomes of this study can not only facilitate understanding of the complexity of the *bZIP* gene family, but also provide novel insight into their crucial roles in abiotic stress responses and anthocyanin biosynthesis in radish.

## 4. Materials and Methods

### 4.1. Identification and Phylogenetic Analysis of the bZIP Gene Family in Radish

The protein annotation file was retrieved from Radish Genome Database (RGD, http://radish-genome.org/) [32]. The candidate *bZIP* genes that included at least one bZIP domain (PF00170, PF07716 or PF03131) were identified using the Pfam protein family database (http://pfam.sanger.ac.uk/) and HMMER 3.0 software (http://hmmer.janelia.org/) [33]. Subsequently, InterPro (http://www.ebi.ac.uk/interpro/) [34] and SMART (http://smart.embl-heidelberg.de/) [35,36] programs were performed to confirm the integrity of the bZIP domain.

All of the *Arabidopsis* bZIP protein sequences [1] were obtained from TAIR database (https://www.arabidopsis.org/). The aa sequences of radish and *Arabidopsis* bZIPs (Appendix A) were imported into MEGA X and multiple sequence alignments were performed using MUSCLE [37]. Thereafter, the alignment file was used to construct NJ phylogenetic tree with following parameters: p-distance, pairwise deletion and bootstrap (1000 replicates) [37,38]. The NJ tree was visualized using Evolview v2 (http://www.evolgenius.info/evolview/) [39]. The phylogenetic tree of all radish bZIP family members was also generated by MEGA X with the same parameters.

### 4.2. Synteny Analysis and Chromosomal Localization

BLASTP search was executed against all potential anchors (E < 1×10^-5^) within the radish genome and between the radish and *Arabidopsis* genomes. Collinear blocks were evaluated using MCScanX software, and alignments with ≤ 1 × 10^-10^ were considered as significant matches [40]. Chromosomal position from RGD and duplications (segmental and tandem) of the *RsbZIP* genes were displayed by TBtools software (https://github.com/CJ-Chen/TBtools).

### 4.3. Analyses of Gene Structure, Conserved Motif, Promoter and GO Annotation

The exon-intron organizations for *RsbZIP* genes were illustrated with Gene Structure Display Server (http://gsds.cbi.pku.edu.cn/) [41]. Identification of conserved motifs was performed by the MEME program (http://meme.nbcr.net/meme/cgi-bin/meme.cgi) [42]. GO annotation of RsbZIP protein sequences was performed using Blast2GO (http://www.blast2go.com). The upstream 1.5 kb genomic sequences of radish *bZIP* genes were submitted to the PlantCARE database (http://bioinformatics.psb.ugent.be/webtools/plantcare/html/) to identify the putative *cis*-elements [43].

### 4.4. Transcriptome-Based Expression Profiling of RsbZIP Genes

The published transcriptome data [18] were used for expression pattern analysis of global *RsbZIP*s. The expression level for each *RsbZIP* gene in leaf and taproot (cortical, cambium, xylem and root tip) at 60 days after germination (DAG) was calculated using the RPKM. Heatmap of *RsbZIP* gene expression profile was generated by TBtools software (https://github.com/CJ-Chen/TBtools). Expression patterns of *RsbZIP* genes under Cd, Cr, Pb, heat and salt stresses were extracted from transcriptome data of radish taproot [19,20,21,22,23].

### 4.5. Plant Materials and Treatments

The radish seeds of ‘NAU-RG’ were sown in the pots containing mixture of isovolumetric soil and substrate, and grown under a 14-h photoperiod at approximately 25 °C/16 °C (day/night) in the growth chamber. Twenty-day-old seedlings were subjected to different stress conditions. The Cd, Cr and Pb treatments were carried out by soaking the 200 mg/L CdCl_2_·2.5H_2_O for 12 h, 600 mg/L K_2_Cr_2_O_7_ for 72 h and 1000 mg/L Pb(NO_3_)_2_ for 72 h [19,20,21]. Heat stress was simulated by 40 ℃ for 24 h in growth chamber [22] and high-salinity stress was imposed by irrigating with 200 mM NaCl for 48 h [23]. After different treatments, the treated roots in each stress were harvested separately for total RNA isolation.

To characterize the potential function of radish *bZIP* genes in anthocyanin biosynthesis, five red-skin (R1, R2, R3, R4, R5) and five white-skin (W1, W2, W3, W4, W5) genotypes were selected and grown in the growth chamber. The root skins of all 10 radish genotypes were harvested at 40 DAG. Additionally, the seeds of one red-skin genotype ‘R1′ were vernalized at 4 ℃ for 10 days and grown until flowering and podding. Six tissue (root skin, flesh, stem, leaf vein, sepal and petal) samples were separately collected, which were frozen immediately in liquid nitrogen and stored at −80 ℃ for further use. For each treatment or tissue, three independent replicates were sampled.

### 4.6. Determination of Anthocyanin Content

The anthocyanins from all above-mentioned samples were extracted in accordance with the method described by Guo et al. [44]. In brief, the supernatant was filtered through 0.22-μm filters for anthocyanin content analysis. The pH differential spectrophotometry method was used to determine total anthocyanin contents in different tissues [44]. Measurements of all samples were replicated three times.

### 4.7. RT-qPCR Analysis

Total RNA extraction was performed with RNAprep Pure Plant Kit (Tiangen, Beijing, China) for RT-qPCR analysis. First-strand cDNA was synthesized using PrimeScript™ RT reagent kit with gDNA eraser (Takara, Dalian, China). Gene expression profiles were generated in triplicate employing a RT-qPCR approach on a LightCycler^®^ 480 System (Roche, Mannheim, Germany) according to the manufacturer’s instructions. The specific primers were listed in Appendix A. The relative expression level was normalized to the *RsActin* gene and calculated by the 2^^−ΔΔC^T^ method [45].

## Figures and Tables

**Figure 1 ijms-20-06334-f001:**
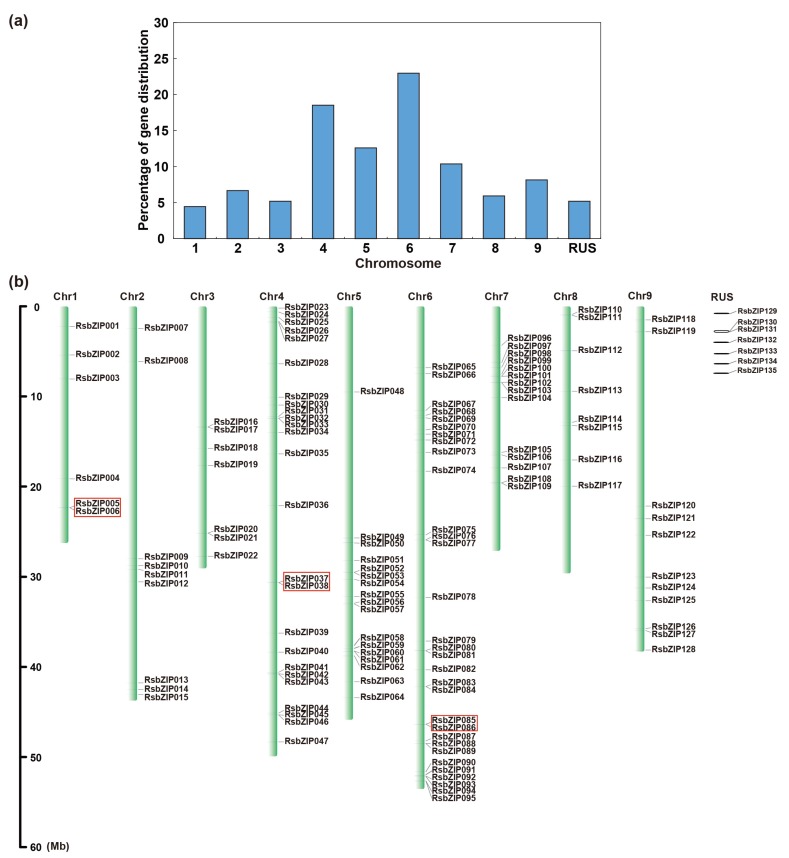
Chromosome localization and distribution of duplication events of radish *bZIP* genes. (**a**) Percentage of *bZIP* genes on each radish chromosome. (**b**) The *RsbZIP* genes were localized onto radish chromosomes (Chr1–9). Tandem duplicated genes on a particular chromosome are indicated in the box. RUS represents radish unassigned scaffolds.

**Figure 2 ijms-20-06334-f002:**
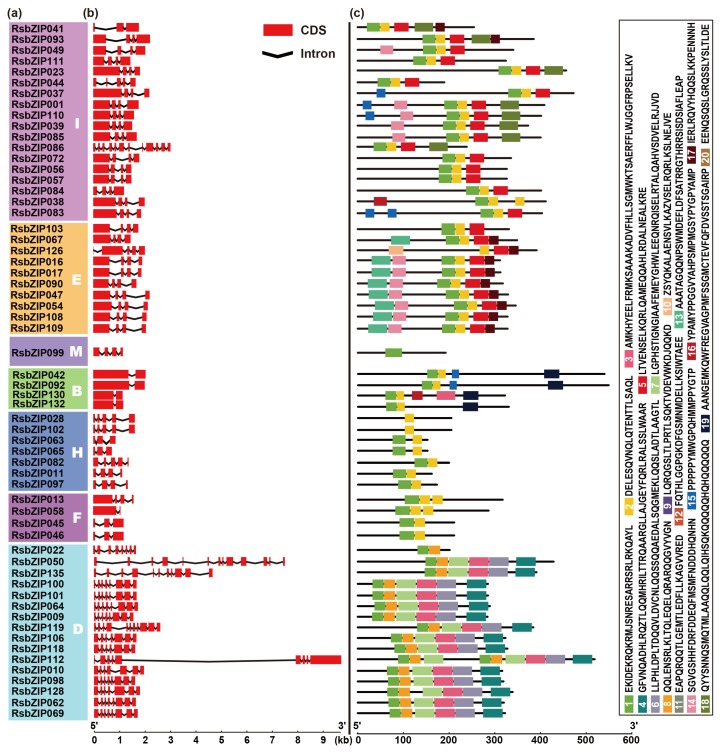
Characterization of radish *bZIP* genes. (**a**) The clustering of RsbZIP proteins based on Neighbor-Joining phylogenetic tree. (**b**) Exon-intron structures of *RsbZIP* genes. Red box and black line indicate exon and intron, respectively. (**c**) Distribution of conserved motifs for RsbZIP proteins.

**Figure 3 ijms-20-06334-f003:**
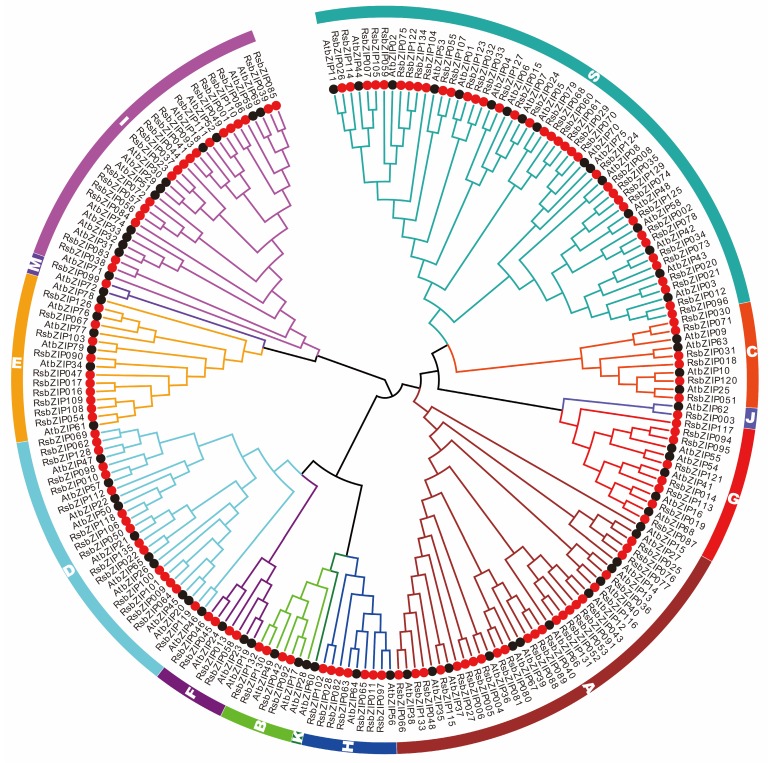
Phylogenetic tree of bZIP proteins from radish (RsbZIP) and *Arabidopsis* (AtbZIP). The colored branches and outer races indicate different subgroups, and the red solid and black circles indicate RsbZIP and AtbZIP proteins, respectively.

**Figure 4 ijms-20-06334-f004:**
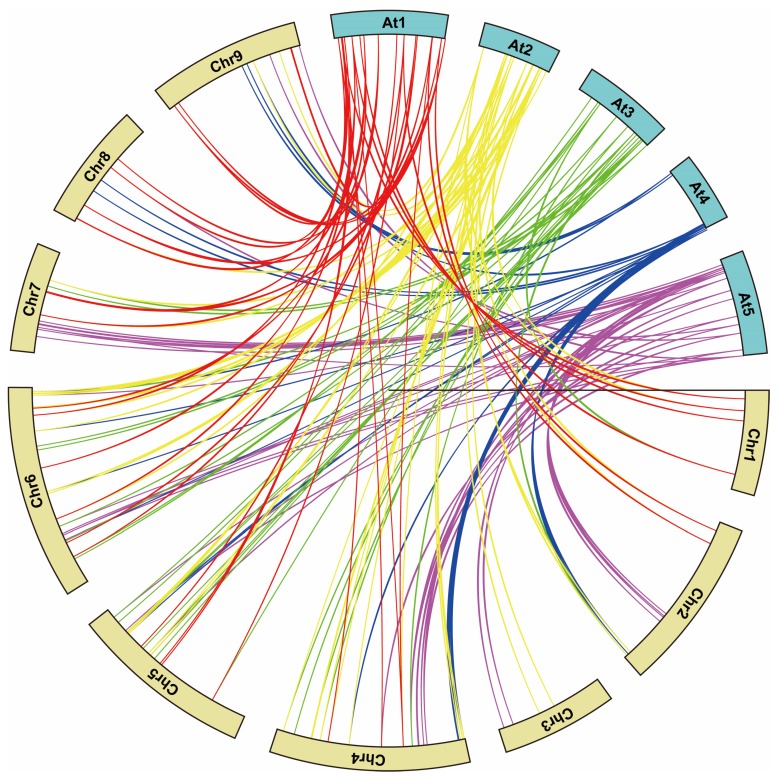
Synteny analysis of *bZIP* genes between radish and *Arabidopsis*. Colored lines connecting two chromosomal regions indicate syntenic regions between radish (Chr1–9) and *Arabidopsis* (At1–5) chromosomes.

**Figure 5 ijms-20-06334-f005:**
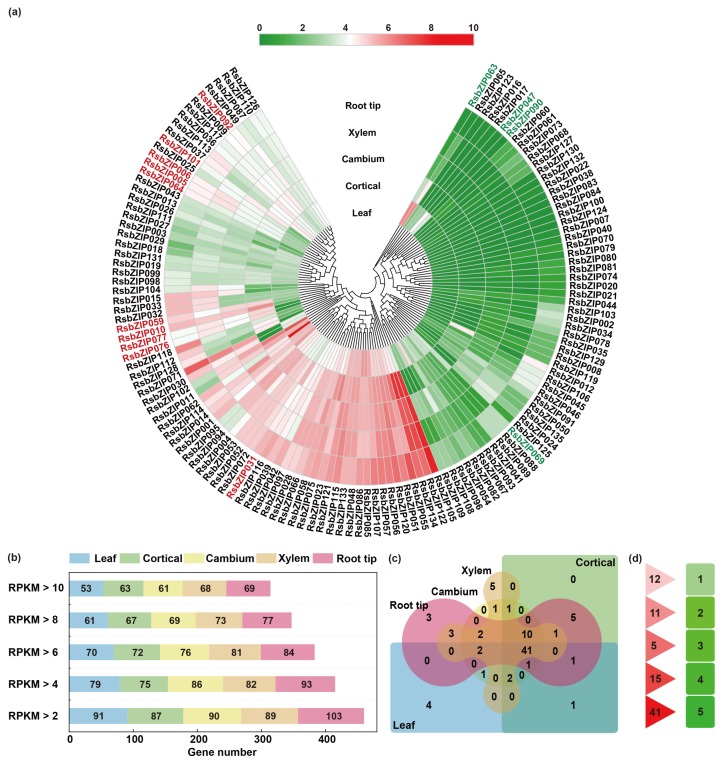
Expression profiles of *RsbZIP* genes in different tissues of radish. (**a**) Heatmap of expression levels for *RsbZIP* genes in five tissues (leaf, cortical, cambium, xylem and root tip). The heatmap was constructed based on the RPKM values of *RsbZIP*s. Changes in gene expression are shown in color as the scale. Red and green fonts indicate the selectively expressed genes with high-levels (RPKM>10) in taproot (cortical, cambium, xylem and root tip) and leaf, respectively. (**b**) The number of *RsbZIP* genes with transcriptional abundance (RPKM>10) in each tissue. (**c**) Venn diagram of overlapping *RsbZIP*s that are abundantly expressed in different tissues. (**d**) Number of *RsbZIP* genes that exhibited abundant expression in 1, 2, 3, 4 and 5 tissues.

**Figure 6 ijms-20-06334-f006:**
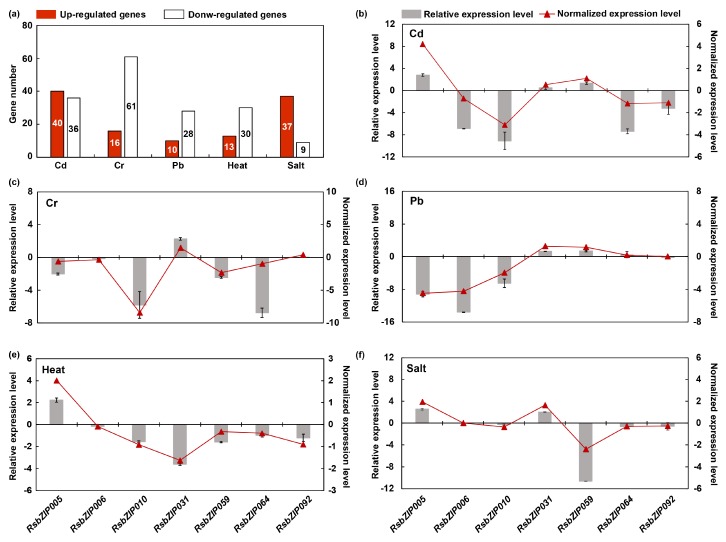
Expression profiles of *RsbZIP* genes under Cd, Cr, Pb, heat and salt stresses. (**a**) The number of *RsbZIP* genes that are up- and down-regulated (the absolute value of Fold change > 1, *p*-value < 0.05) under Cd, Cr, Pb, heat and salt stresses. (**b**–**f**) Expression profiles of seven *RsbZIP* genes that specifically high-expressed in taproot under Cd, Cr, Pb, heat and salt stresses from RNA-seq data and RT-qPCR analysis. Red bar represents transcript abundance change calculated by the RPKM method. Gray bar with associated standard error bar indicates relative expression level determined by RT-qPCR analysis.

**Figure 7 ijms-20-06334-f007:**
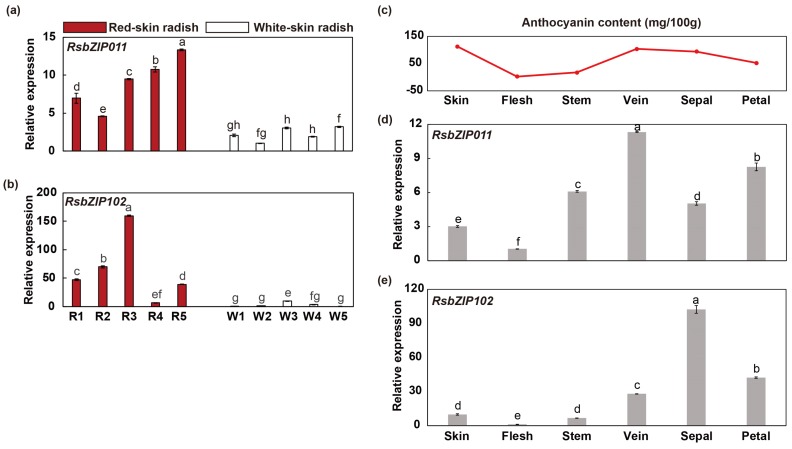
Expression profiles of *RsbZIP* genes potentially involved in anthocyanin biosynthesis pathway. Here, (**a**) and (**b**) show the expression profiles of *RsbZIP011* and *RsbZIP102* in five red-skin (R1, R2, R3, R4, R5) and five white-skin radish genotypes (W1, W2, W3, W4, W5). (**c**) Anthocyanin content in six different tissues (skin, flesh, stem. vein, sepal and petal) of radish. Moreover, (**d**) and (**e**) Expression profiles of *RsbZIP011* and *RsbZIP102* in six different tissues. Error bars indicate standard deviation based on three replicates. Letters represent significant differences at a 0.05 level based on Duncan’s test.

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
