# Peer review of "Genome- and Transcriptome-Wide Characterization of bZIP Gene Family Identifies Potential Members Involved in Abiotic Stress Response and Anthocyanin Biosynthesis in Radish (Raphanus sativus L.)"

_ijms, 2019, doi:10.3390/ijms20246334_

Round 1

Reviewer 1 Report

Manuscript titled: Genome-Wide Characterization of bZIP Gene Family Identifies Potential Members Involved in Abiotic Stress Response and Anthocyanin Biosynthesis in Radish is an interesting and systematic report on the role of basic leucine zipper (bZIP) transcription factors in Raphanus sativus in various abiotic stresses (Cd, Cr, Pb, drought and salt stress) with particular focus to anthocyanin accumulation. It was found that two and six RsbZIPs were significantly co-upregulated and co-downregulated by Cd, Cr and Pb stresses, while 19 and 18 RsbZIP genes were induced by drought and salt stresses, respectively. RT-qPCR analysis revealed that specific RsbZIP011 and RsbZIP102 were significantly up-regulated in the tissues of radish plants with high anthocyanin content. MS is well written. The methodology is adequately described. Authors should improve only some of the figures which are not readable in current form. Thus I recomend this article for publication after minor revision (see document in att.)

Author Response

Response to Editor Comments

Point 1: In addition, we noticed that there is a citation in the figure legend. We are wondering whether this figure is involved with copy right issue? If the answer is yes, we need you to provide copyright permission. If the answer is no, please provide the PDF of the cited article so that we can confirm it isn’t involved with copyright problem. Our intention is to avoid some dissension appeared between you and the copyright owner after your paper is published. Hope you could kindly understand that.

Response 1: Thanks for the reviewer’s helpful comments and suggestions. Figure 3 is not involved with copy right issue. In Figure 3, the bZIP protein sequences and their subgroups names of Arabidopsis in the article from Dröge-Laser et al. (2018) (https://doi.org/10.1016/j.pbi.2018.05.001) was used for the phylogenetic analysis of radish and Arabidopsis.

Response to Reviewer 1 Comments

Point 1: English language and style are fine/minor spell check required.

Response 1: Thanks for the reviewer’s constructive suggestion. The language of this paper had been comprehensively polished by a native English-speaking colleague. Hopefully it’s more acceptable for publication on this journal.

Point 2: ‘Among them, 128 of 135 RsbZIP members were localized on the radish chromosomes (Chrs) of’. Do you miss something here?

Response 2: Thanks for the reviewer’s constructive suggestion. We re-checked this sentence and found that “radish” is missed here. In the revised manuscript, this sentence was revised as “Among them, 128 of 135 RsbZIP members were localized on the nine chromosomes (Chrs) of radish” (Page 2, Line 83).

Point 3: Figure 2 is not readable. I understand that it is huge figure, and readers may see details only if they zoom in figure. So please editor to decide is that acceptable.

Response 3: Thanks for the reviewer’s helpful suggestion. Now, we provide a more clear and readable Figure 2 in this revised manuscript (Page 4-5).

Point 4: Legend of figure 5 should be below figure. Fonts on figures are too tiny so I would suggest authors to enlarge what is possible.

Response 4: Thanks for the reviewer’s helpful suggestion. We have moved the legend to the bottom of Figure 5, and the fonts on figures have been enlarged according to the reviewer’s comment (Page 8).

Reviewer 2 Report

The paper describes the systematic bioinformatic analysis (genome and transcriptomics-wide) of bZIP Basic leucine zipper transcription factors in Radish (Raphanus sativus L.).

Meanwhile most of the data related to RNAseq were generated by the authors (and they point to their own publications related to metal, heat and drought) other RNAseq data on Radish were extrapolated from public transcriptomics data coming from Japanese authors ( Mitsui, Yet al. Sci. Rep. 2017).

Despite the authors keep saying their represent the first genome-wide analysis of bZIP, no genomics studies are contemplated in this manuscript. The bioinformatics analysis of the genes has been possible by accessing public genome data of Radish coming from the Raphanus sativus Genome DataBase (http://radish.kazusa.or.jp/).

Furthermore, no correct acknowledgement is given to the authors that have made a draft of the Radish genome. Hence the introduction should contain a brief info on Radish as plant itself and common use (importance as stable food). The fact that the first genome draft has been published and because of that the proteins/mRNA transcriptomics data are available is also a big push forward the development to this analysis of the bZIP TFs.

The title should contain botanical info on Radish and should be hence changed in:

"Genome and Transcriptomics-Wide Characterization of bZIP Gene Family Identifies Potential Members Involved in Abiotic Stress Response and Anthocyanin Biosynthesis in Radish (Raphanus sativus L.)".

In the abstract only a number of genes involved in abiotic stresses analysed here are reported. In the conclusions no clear pointing on which candidates are responsible for each abiotic stress. In the lanes 289-294 the description of which candidate bZIPs related to the heavy metals, heat and salt stress is quantitative but not qualitative. It seems that the authors are grouping the all of these abiotic stresses into a big container (as the Figure 6 shows).

Taken apart (and not together) the authors should analyse each metal for the heavy metal stress, heat and salt/drought stress and propose a narrowed down candidate pool for each. It seems also that the qPCR validation for heat stress and heavy metal is not performed.

Meanwhile the anthocyanin inducible bZIP are well described and candidates have been clearly identified matching homologous genes in other species. Furthermore, the validation of the RsbZIP011 and RsbZIP102 has been well executed and it seems robust.

The authors need to narrow down the RsbZIP genes in taproot, and provided more qPCR evidence for their roles in response to Cd, Cr, Pb and heat stresses. No clear small pool of candidates have been pointed out for these abiotic stresses and the reader is confused after looking at figure 6. Else they should remove the abiotic stress related to Cd, Cr, Pb and heat stresses from this manuscript.

The figure 2 is also very small to read.

Author Response

Response to Reviewer 2 Comments

Point 1: The paper describes the systematic bioinformatic analysis (genome and transcriptomics-wide) of bZIP Basic leucine zipper transcription factors in Radish (Raphanus sativus L.). Meanwhile most of the data related to RNAseq were generated by the authors (and they point to their own publications related to metal, heat and drought) other RNAseq data on Radish were extrapolated from public transcriptomics data coming from Japanese authors (Mitsui, Y et al. Sci. Rep. 2017). Despite the authors keep saying their represent the first genome-wide analysis of bZIP, no genomics studies are contemplated in this manuscript. The bioinformatics analysis of the genes has been possible by accessing public genome data of Radish coming from the Raphanus sativus Genome DataBase (http://radish.kazusa.or.jp/).

Furthermore, no correct acknowledgement is given to the authors that have made a draft of the Radish genome. Hence the introduction should contain a brief info on Radish as plant itself and common use (importance as stable food). The fact that the first genome draft has been published and because of that the proteins/mRNA transcriptomics data are available is also a big push forward the development to this analysis of the bZIP TFs.

Response 1: Many thanks for the reviewer’s constructive suggestions. As we know, the first draft sequences of the radish genome were released into the Raphanus sativus Genome DataBase (http://radish.kazusa.or.jp/) in 2014, which provided valuable and useful genome basis for systematic identification of functional genes and TFs associated with important horticultural traits in radish.

At the beginning of this study, we try to isolate and identify the bZIP TFs from the Raphanus sativus Genome DataBase (Kitashiba et al., 2014) and some bZIPs were successfully identified, indicating the proteins/mRNA transcriptomics data are available to perform the genome-wide analysis of the TFs in radish. However, due to the fact that only 116.0 Mb scaffolds (21.8% of the estimated genome) were assigned to the pseudo-chromosomes in this genome version, it’s hard to carry out the precise chromosome distribution of radish bZIP genes and consequently synteny analysis of bZIP genes between radish and Arabidopsis. Afterwards, a chromosome-level radish genome database (http://radish-genome.org/) was published (Jeong et al., 2016), which could meet our requirement to conduct the identification and chromosome distribution of radish bZIP genes. Therefore, we used this newly radish genome database to identify the bZIP TFs.

In this revised manuscript, we added the brief information on radish as stable food and emphasized the importance of the first radish genome draft in introduction according to the reviewer’s suggestions as follows.

Firstly, “Radish (Raphanus sativus L.), one of the most widely cultivated root vegetable crops, is a great source of phytochemicals such as anthocyanins.” was revised as Radish (Raphanus sativus L.), belonging to the Brassicaceae family, is a major root vegetable crop with plenty of nutrition, and its taproot is an edible and stable food supplying a great source of phytochemicals such as anthocyanins [16,17]. in introduction (Page 2, Lines 57-59).

Secondly, “The first draft sequences of the radish genome were released into the Raphanus sativus Genome DataBase (http://radish.kazusa.or.jp/) in 2014 [16], and its available genomic sequences and proteins/mRNA transcriptomics data provided valuable information for systematic identification of functional genes and TFs associated with important horticultural traits in radish.” was added in introduction (Page 2, Lines 63-67).

Thirdly, the sentence ‘this study represents the first genome-wide characterization of bZIP gene family in radish.’ was revised as ‘this study describes the systematic genome and transcriptome-wide characterization of bZIP gene family in radish.’ (Page 12, Lines 328-329).

Additionally, the literature “Draft Sequences of the Radish (Raphanus sativus L.) Genome” (Kitashiba et al., 2014) was cited in our manuscript (Page 15, Lines 460-463).

Point 2: The title should contain botanical info on Radish and should be hence changed in:

"Genome and Transcriptomics-Wide Characterization of bZIP Gene Family Identifies Potential Members Involved in Abiotic Stress Response and Anthocyanin Biosynthesis in Radish (Raphanus sativus L.)".

Response 2: Thanks for the reviewer’s constructive suggestion. We have changed the title into “Genome and Transcriptome-Wide Characterization of bZIP Gene Family Identifies Potential Members Involved in Abiotic Stress Response and Anthocyanin Biosynthesis in Radish (Raphanus sativus L.)” (Page 1, Lines 2-6).

Point 3: The authors need to narrow down the RsbZIP genes in taproot, and provided more qPCR evidence for their roles in response to Cd, Cr, Pb and heat stresses. No clear small pool of candidates have been pointed out for these abiotic stresses and the reader is confused after looking at figure 6. Else they should remove the abiotic stress related to Cd, Cr, Pb and heat stresses from this manuscript.

Taken apart (and not together) the authors should analyse each metal for the heavy metal stress, heat and salt/drought stress and propose a narrowed down candidate pool for each. It seems also that the qPCR validation for heat stress and heavy metal is not performed.

Response 3: Thanks for the reviewer’s constructive suggestion. Firstly, we found that a total of ten RsbZIP genes (RsbZIP005/006/010/031/059/064/076/077/092/101) exhibited high expression level in the taproot (RPKM>10) from the previous public transcriptomics data (Mitsui et al., 2015) (Pages 7-8, Lines 169-173, Figure 5a). Then, to explore whether these high-expressed RsbZIP genes were involved in different abiotic stresses in the taproot, the RT-qPCR was performed to analyses the expression profiles of these ten RsbZIP genes under Cd, Cr, Pb, heat and salt stresses according to the reviewer’s suggestion.

Accordingly, “As shown in Figure 6a, a total of 36, 44 and 24 RsbZIP genes were differentially expressed under Cd, Cr, Pb stresses……RsbZIP genes were differentially expressed under salt stress (Figure 6a).” was revised as follow:

As shown in Figure 6b-f, seven of these ten RsbZIP genes exhibited significantly differential expression in taproot under the Cd, Cr, Pb, heat and salt treatments. For example, the RsbZIP010 gene exhibited down-regulated expression under three heavy metal stresses, whereas the RsbZIP005 gene was significantly up- and down-regulated under Cd and Cr stress, respectively, indicating that the RsbZIP010 gene might play conserved key roles in the biological process of response to three heavy metal stress in radish taproot. Notably, the RsbZIP031 and RsbZIP059 gene showed significantly down-regulation under heat and salt stresses, respectively. This finding indicated that these two genes may act as specific and critical regulators in heat and salt stress response in radish taproot (Page 9, Lines 190-198).

Point 4: In the abstract only a number of genes involved in abiotic stresses analysed here are reported. In the conclusions no clear pointing on which candidates are responsible for each abiotic stress. In the lanes 289-294 the description of which candidate bZIPs related to the heavy metals, heat and salt stress is quantitative but not qualitative. It seems that the authors are grouping the all of these abiotic stresses into a big container (as the Figure 6 shows).

Response 4: Thanks for the reviewer’s constructive suggestions. Based on the results about2.5 Potential Roles of RsbZIP Genes in Various Abiotic Stress Responses’, we updated the abstract and conclusions according to the reviewer’s suggestion. Hopefully it could make more sense on pointing out potential candidates responsible for each abiotic stress.

In the abstract section,

“Expression profiling revealed that two and six RsbZIPs were significantly co-upregulated and co-downregulated by Cd, Cr and Pb stresses.” was revised as “Transcriptome analysis revealed that ten RsbZIP genes exhibited high-expression levels in radish taproot (RPKM > 10). Specifically, RsbZIP010 exhibited down-regulated expression under Cd, Cr and Pb stresses, whereas RsbZIP031 and RsbZIP059 showed significantly down-regulation under heat and salt stresses, respectively.” (Page 1, Lines 17-20)

In the conclusions section,

“Transcriptomic and RT-qPCR analysis revealed diverse expression profiles of RsbZIP genes in taproot, and provided evidence for their roles in response to Cd, Cr, Pb, heat, salt and drought stresses.” was revised as “Transcriptomic and RT-qPCR analysis revealed diverse expression profiles of RsbZIP genes in taproot. Notably, the RsbZIP010 gene exhibited down-regulated expression under Cd, Cr and Pb stresses, indicating that it might play conserved key roles in the biological process of response to three heavy metal stresses in radish taproot.” (Page 12, Lines 330-333)

Accordingly, Figure 6 was also updated (Page 9) according to the reviewer’s comments.

Point 5: The figure 2 is also very small to read.

Response 5: Thanks for the reviewer’s helpful suggestion. Now, we provide a more clear and readable Figure 2 in this revised manuscript (Pages 4-5) according to the reviewer’s suggestions.

Round 2

Reviewer 2 Report

Well done, all the suggested adjustments have been performed. All the figure read better and in the text the wanted

changes have been made.

Only small changes are needed:

Lane 83:  add to "HMM profile" "analysis", since it is an analysis before being a profile.

Lane 218: instead of "evaluate" you better say "validate", because you actually use qPCR to validate the already known RNAseq data, which you have already "evaluated".

Lane 336: the correct usage is "Especially" but in this context is better to use, "In particular". 

 From lane 374: In the summary/conclusions. It is better to add that except the heat stress, from fig. 6 is evident that RsbZIP031 is instead upregulated in all the metal stresses considered. But you need to emphasize this concept in the summary.

After lane 339 add something like: "However, the RsbZIP031 was instead upregulated in all the metal stresses considered (Fig.6) showing its involvement maybe into salt stress and heavy metal detoxification, that in the specific case of Pb and Cd stresses the RsbZIP059 was also upregulated". 

Author Response

Response to Reviewer 2 Comments

Point 1: Lane 83: add to "HMM profile" "analysis", since it is an analysis before being a profile.

Response 1: Many thanks for the reviewer’s helpful suggestion. In this revised manuscript, we have changed “HMM profile” into “HMM profile analysis” (Now in Page 2, Line 80).

Point 2: Lane 218: instead of "evaluate" you better say "validate", because you actually use qPCR to validate the already known RNAseq data, which you have already "evaluated".

Response 2: Thanks for the reviewer’s helpful suggestion. We replaced “evaluate” with “validate” (Now in Page 9, Line 200).

Point 3: Lane 336: the correct usage is "Especially" but in this context is better to use, "In particular".

Response 3: Thanks for the reviewer’s helpful comment. Now, the “especially” was revised as “in particular” (Now in Page 11, Line 294).

‘….and stress-related cis-elements were found in the promoter regions of most RsbZIP genes, especially the ABRE and ERE elements, which…..’ was revised as …….. and stress-related cis-elements including the ABRE and ERE were found in the promoter regions of most RsbZIP genes, which….’ (Now in Page 12, Lines 302-303)

Point 4: From lane 374: In the summary/conclusions. It is better to add that except the heat stress, from fig. 6 is evident that RsbZIP031 is upregulated in all the metal stresses considered. But you need to emphasize this concept in the summary.

Response 4: Thanks for the reviewer’s constructive suggestion. We have added the sentence “Except the heat stress, the RsbZIP031 gene was up-regulated under salt and three heavy metal (HM) stresses, suggesting it might act as important regulator in the regulation of salt and HM stress responses.” in the summary/conclusions (Now in Page 12, Lines 336-338).

Point 5: After lane 339 add something like: "However, the RsbZIP031 was instead upregulated in all the metal stresses considered (Fig.6) showing its involvement maybe into salt stress and heavy metal detoxification, that in the specific case of Pb and Cd stresses the RsbZIP059 was also upregulated".

Response 5: Thanks for the reviewer’s constructive comment. We have added the sentence “However, RsbZIP031 was up-regulated under salt and three HM stresses, showing its involvement in response to salt stress and HM detoxification. Moreover, the RsbZIP059 was up-regulated under Pb and Cd stresses.” in the discussion, according to the reviewer’s suggestion (Now in Page 11-12, Lines 296-298).

‘….and stress-related cis-elements were found in the promoter regions of most RsbZIP genes, especially the ABRE and ERE elements, which…..’ was revised as …….. and stress-related cis-elements including the ABRE and ERE were found in the promoter regions of most RsbZIP genes, which….’ (Now in Page 12, Lines 302-303)